# Transcriptome Analysis Reveals Mechanisms of Stripe Rust Response in Wheat Cultivar Anmai1350

**DOI:** 10.3390/ijms26125538

**Published:** 2025-06-10

**Authors:** Feng Gao, Jingyi Zhu, Xin Xue, Hongqi Chen, Xiaojin Nong, Chunling Yang, Weimin Shen, Pengfei Gan

**Affiliations:** 1Anyang Academy of Agriculture Sciences, Anyang 455000, China; gaofeng5182@163.com (F.G.); futurenini@163.com (X.X.); aynky123@yeah.net (H.C.); futurexx@126.com (C.Y.); gaofeng5182@126.com (W.S.); 2State Key Laboratory of Crop Stress Resistance and High-Efciency Production, College of Plant Protection, Northwest A&F University, Xianyang 712100, China; nwafu_zjy@163.com (J.Z.); nongxiaojin@nwafu.edu.cn (X.N.)

**Keywords:** Anmai1350, *Puccinia striiformis* f. sp. *tritici*, RNA-seq, reactive oxygen species, plant–pathogen interactions

## Abstract

Wheat (*Triticum aestivum* L.) is the world’s most indispensable staple crop and a vital source of food for human diet. Wheat stripe rust, caused by *Puccinia striiformis* f. sp. *tritici* (*Pst*), constitutes a severe threat to wheat production and in severe cases, the crop fails completely. Anmai1350 (AM1350) is moderately resistant to leaf rust and powdery mildew, and highly susceptible to sheath blight and fusarium head blight. We found that the length and area of mycelium in AM1350 cells varied at different time points of *Pst* infection. To investigate the molecular mechanism of AM1350 resistance to *Pst*, we performed transcriptome sequencing (RNA-seq). In this study, we analyzed the transcriptomic changes of the seedling leaves of AM1350 at different stages of *Pst* infection at 0 h post-infection (hpi), 6 hpi, 24 hpi, 48 hpi, 72 hpi, and 120 hpi through RNA-seq. Quantitative Real-Time Polymerase Chain Reaction (qRT-PCR) was used to validate RNA-seq data. It was determined that there were differences in the differentially expressed genes (DEGs) of AM1350, and the upregulation and downregulation of the DEGs changed with the time of infection. At different time points, there were varying degrees of enrichment in the response pathways of AM1350, such as the ”MAPK signaling pathway–plant”, the “plant–pathogen interaction” pathway and other pathways. After *Pst* infected AM1350, the reactive oxygen species (ROS) content gradually increases. The ROS is toxic to *Pst*, promotes the synthesis of phytoalexins, and inhibits the spread of *Pst*. As a result, AM1350 shows resistance to *Pst* race CYR34. The main objective of this study is to provide a better understanding for resistance mechanisms of wheat in response to *Pst* infections and to avoid production loss.

## 1. Introduction

Wheat is one of the most valued crops playing an important role in global food security, providing approximately 20% of calories and protein in the human diet [1,2]. Previously reported yellow rust resistance genes have already lost effectiveness and caused yield loss by the emergence of new *Puccinia striiformis* f. sp. *tritici* (*Pst*) races [3]. A new *Pst* race (V26/CYR34) was virulent to the resistance gene *Yr26* (*Yellow rust resistance 26*) and spread in wheat breeding programs in many major wheat-planting regions [4,5]. About 82% of the varieties are susceptible to one or more races of *Pst* (CYR32, CYR33 and CYR34). Stripe rust mainly overwinters in the overwintering regions, and the wheat materials in the overwintering regions are vulnerable to the *Pst* race CYR34 race [6,7,8,9]. Thus, prior to *Pst* race CYR34 inflicting greater losses on wheat production, it is of utmost importance to develop new resistant cultivars or gain an understanding of the defense response mechanism of wheat against *Pst*.

The molecular mechanisms underlying the interaction between wheat and *Pst* can be deciphered through global gene expression profiling approaches such as RNA sequencing (RNA-seq). RNA-seq has been widely applied to investigate the mechanisms involved in the interaction between wheat and pathogenic fungi [10,11,12,13,14]. However, resistance mechanisms can be race-specific and can be dependent on the resistance mechanisms of host plant. Despite of wide ranges of molecular studies different host plants, the resistance mechanisms of the AM1350 against CYR34 of *Pst* is unclear. Numerous studies have indicated that the photosynthetic activity has a close connection with wheat’s response to *Pst*. The infection of *Pst* can readily impair the photosystem II of wheat, thereby leading to a decrease in photosynthetic capacity [9,15,16,17]. Moreover, during the *Pst* infection process, the antioxidant enzyme activities and chlorophyll fluorescence in stripe rust-resistant wheat were found to be higher compared to those in susceptible wheat [18,19]. When contrasted with plants having smaller genomes, the application of most genetic and molecular techniques for studying the genes involved in the wheat-*Pst* interaction has been restricted. This is attributed to the large and complex genome of hexaploid wheat, which makes transformation arduous [20,21]. Additionally, fungi exhibit sexual reproduction, and genes that are not essential for biotrophy can experience irreversible deletion.

To deepen our understanding of the resistance mechanisms of wheat in response to *Pst* infection and improve wheat yield, we selected Anmai1350 (AM1350) for transcriptome analysis to investigate the phenotypic responses and co-regulated mRNA transcriptomic responses in wheat. This material exhibits changes in expression patterns after infection with the *Pst* race CYR34, aiming to identify differentially expressed genes (DEGs) and signal pathways related to the response to *Pst*. This study provides data resources for analyzing the mechanism of wheat resistance to stripe rust.

## 2. Results

### 2.1. Changes in Mycelial Length and Area During the Infection Process

When both MX169 and AM1350 were inoculated with *Pst* race CYR34, the spore production on the leaf surface of AM1350 decreased significantly (Figure 1A), demonstrating that AM1350 has the characteristic of resistance to *Pst*. The thallus of *Pst* is a mycelium, which is filamentous and septate, spreading between host cells. In contrast, the mycelia in the central part of the colony are densely arranged, with the thallus swelling, deforming, and becoming thick, and irregular protrusions occurring in some parts [21]. The growth and development of *Pst* on AM1350 and MX169 plants inoculated with *Pst* race CYR34 were observed. The *Pst* was subjected to staining with Wheat Germ Agglutinin (WGA) and subsequently examined under a fluorescence microscope (Figure 1B,C). The length and area of the mycelia change over time during the *Pst* infection process (Figure 1D,E). From 24 hpi to 48 hpi, there were no significant changes in either the length or the area of the mycelia. As time progresses, the length and area of the mycelia increase during the *Pst* infection process.

### 2.2. Statistical Analysis of RNA-seq Results from Different Time Points After Pst Inoculation

As shown in Table 1, a total of 934,632,590 raw reads were produced from the 21 libraries. After filtering out low-quality reads, adapter sequences, and reads containing poly-N, a set of clean reads was obtained. The values of “Q20” and “Q30” were over 97% and 93% respectively, and the GC content varied from 53.80% to 56.27% (Table 1). These statistical figures imply that the sequencing data were of high quality. Subsequently, the clean reads were mapped to the reference genome, with an average mapping rate ranging from approximately 92.70% to 93.55%. These results illustrate that the sequencing and alignment processes were highly reproducible and reliable.

### 2.3. Characterization of DEGs in Response to Pst Infection

Based on the criteria of |log2Foldchange| ≥ |1|, Padj < 0.05, and FPKM ≥ 1, 24,989, 24,023, 20,927, 11,505, 12,626, and 8233 DEGs were obtained at different hpi (6 hpi, 12 hpi, 24 hpi, 48 hpi, 72 hpi, and 120 hpi), respectively. Among them, 11,271, 10,131, 8875, 5408, 5282, and 2614 DEGs were upregulated, while 13,718, 13,892, 12,052, 6097, 7344, and 5619 DEGs were downregulated (Figure 2A). Venn diagram analysis was utilized to identify the common and unique genes present in different control groups. At 6 hpi, there were 4314 unique genes expressed, and at 120 hpi, there were 546 unique genes expressed (Figure 2B). At 6 hpi, there were 2105 unique genes with downregulated expression, and at 120 hpi, there were 323 unique genes with downregulated expression (Figure 2C). At 6 hpi, there were 3169 unique genes with upregulated expression, and at 120 hpi, there were 450 unique genes with upregulated expression (Figure 2D).

The gene expression response of AM1350 shows dynamic changes at different stages after infection by *Pst*. The changes in the upregulation and downregulation of DEGs reflect regulatory activities such as activation and inhibition of wheat genes during the infection process, and there are differences in the rate of change in wheat gene expression. The expression of DEGs and their upregulation and downregulation in AM1350 after the initial infection by *Pst* are both higher than those of the gene expression in response of wheat in the later stage.

### 2.4. Validation of RNA-seq Data by qRT-PCR

To accurately evaluate the accuracy of RNA sequencing data, four DEGs were selected for expression analysis by qRT-PCR (Figure 3A–D). *TaATPD* (ATP synthase delta chain, Gene ID: TraesCS4D02G066200) showed a downward trend in expression from 6–24 h, with levels comparable to those at 0 h from 48–120 h; *TaICL* (Isocitrate lyase, Gene ID: TraesCS2B02G244600) exhibited high expression from 6 hpi to 24 h, followed by a decrease to levels similar to the 0 h sample; *TaWRKY24* (WRKY transcription factor WRKY24, Gene ID: TraesCS1D02G072900) displayed no significant expression differences at 6 hpi, 12 hpi, and 24 hpi, but significant downregulation at 48 hpi, 72 hpi, and 120 hpi; *TaTIFY6A* (TIFY domain-containing protein 6a, Gene ID: TraesCS5A02G533100) showed an overall downward expression trend, though with an upward trend at 48 hpi and 72 hpi compared to 6 hpi, 12 hpi, and 24 hpi. The results showed that although the fold changes of genes showed more or less differences compared with the transcriptome data, the qRT-PCR results of these DEGs were consistent with the trend of transcriptome data changes (Figure 3E). This indicates that the RNA sequencing results of this study are reliable for various analyses.

### 2.5. Cluster Heatmap Group and KEGG Enrichment Analyses of DEGs

The DEGs from all comparison groups were combined to form a differentially expressed gene set. We employed the mainstream hierarchical clustering method on the FPKM values of genes, and normalized the rows (using Z-score). The color in every cell does not stand for the gene expression value; rather, it is the normalized value achieved by normalizing the rows of the expression data. One can only make horizontal, not vertical, comparisons of the colors in the heatmap [22]. The outcome demonstrates that in the cluster diagram of the genes, the abscissa stands for the sample names, and the ordinate represents the normalized FPKM values of DEGs. The more intense the red color, the higher the expression level, whereas the more vivid the green color, the lower the expression level. The results showed that after AM1350 was infected by *Pst*, the gene expression patterns were similar within 6 h to 24 h, and the gene expression patterns were similar within 48 h to 72 h. Some differences were also observed in the experiment, indicating that there are both similar and different parts in the resistance response mechanisms of AM1350 to *Pst* at different times. That is, different response pathways are adopted at different stages to enhance the resistance of AM1350 to stripe rust. (Figure 4A). To identify the metabolic pathways activated by the stimulus, we searched for DEGs in the KEGG database and obtained the significantly enriched pathways in the control group (Figure 4B). The pathways in AM1350, such as “Peroxisome” and “Glycine, serine, and threonine metabolism” were significantly enriched at 6 hpi. In the “Glycine, serine, and threonine metabolism” pathway, there are 32 DEGs, among which 28.125% are upregulated, and the rest are downregulated. And in the “Peroxisome” pathway, there are 30 differentially expressed genes (DEGs), half of which are upregulated and the other half are downregulated. Pathways including ”Carbon metabolism” and “Plant hormone signal transduction” were significantly enriched at 12 hpi. In the plant hormone signal transduction pathway, there are a total of 63 DEGs, among which 20 DEGs are upregulated, and the rest are downregulated. At 24 hpi, pathways such as “Carbon metabolism” and “Biosynthesis of amino acids” were significantly enriched. *TaATPD* (ATP synthase delta chain, Gene ID: TraesCS4D02G066200) related to “Photosynthesis” pathway exhibited significant downregulation at 6 hpi, 12 hpi, and 24 hpi. The results of qRT-PCR showed that the expression level exhibited a clear downregulation at 6 hpi, 12 hpi, and 24 hpi. However, at 48 hpi, 72 hpi, and 120 hpi, the expression level showed no differences in gene expression at 0 hpi (Figure 3A). *TaICL* (Isocitrate lyase, Gene ID: TraesCS2B02G244600) related to “Carbon metabolism” pathway exhibited significant differential expression at 6 hpi, 12 hpi, and 24 hpi. The results of qRT-PCR showed that it had upregulated at 6 hpi, 12 hpi, and 24 hpi. At 48 hpi, gene expression levels decreased (Figure 3B). *TaWRKY24* (WRKY transcription factor WRKY24, Gene ID: TraesCS1D02G072900) on the “plant–pathogen interaction” pathway exhibited significant differential expression at 48 hpi, 72 hpi, and 120 hpi. The result of qRT-PCR showed downregulation at 24 hpi, 48 hpi, 72 hpi, and 120 hpi (Figure 3C). *TaTIFY6A* (TIFY domain-containing protein 6a, Gene ID: TraesCS5A02G533100) related to the plant hormone signal transduction pathway showed significant differential expression at 6 hpi, 12 hpi, and 24 hpi. There was an upregulation at 48 hpi, 72 hpi, and 120 hpi compared to that of 6 hpi, 12 hpi, and 24 hpi (Figure 3D). The detailed qRT-PCR expression levels along with RNA-seq data are presented in Appendix A. The detailed KEGG enrichment pathways are presented in Appendix A. At 120 hpi, the significantly enriched pathways included the “MAPK signaling pathway–plant”, plant hormone signal transduction, and “plant–pathogen interaction” [23].

### 2.6. Weighted Gene Correlation Network Analysis

WGCNA is a methodology in systems biology aimed at depicting the patterns of gene co-association across various samples. It has the capacity to pinpoint gene sets that exhibit a high degree of coordinated change [24]. The results show that each color corresponds to a module, and every gene on the clustering tree belonging to the same color is part of the same module. After AM1350 is infected by *Pst*, there are differences in the gene expression levels corresponding to different time points. We grouped the time points with similar gene expression patterns into the same module. The DEGs generated in this process share some similarities in their functions. The specific module information is presented (Appendix A Heatmap). The height values of the DEGs in the modules labeled as MEblue and MEcyan are less than 0.75, and the vertical distances are small, which indicates that the correlation among the genes within these modules is high, and their gene expression patterns are similar. Studying the gene expression results at the corresponding time periods within these modules is helpful for understanding the resistance mechanism of AM1350 to *Pst* (Figure 5A). In the module–sample relationship, the abscissa represents the samples, and the ordinate represents the modules. The number in each cell represents the correlation between the module and the sample. As this value approaches 1, the positive correlation between the module and the sample intensifies. In contrast, as it nears −1, the negative correlation grows stronger. The number inside the parentheses stands for the significance *p*-value. The smaller this value is, the stronger the significance. When the correlation between a module and a sample is significantly higher than that of other modules, it indicates that this module may have the strongest association with the sample. It can be easily seen from the figure that there is a strong positive correlation between AM1350_0h and the MEbrown module. AM1350_6h has a strong correlation with the MElightcyan module. Both AM1350_24h and AM1350_12h have a significant correlation with the MEgreenyellow module, which indicates that the transcriptional levels within AM1350 are similar from 12 to 24 h after being infected by *Pst* (Figure 5B). Based on the module–sample relationship and the cluster heatmap of DEGs, we can comprehensively analyze the gene expression profiles of AM1350 at different time points after being infected by the stripe rust fungus, and thus better analyze the resistance mechanisms of AM1350 at different time periods (Figure 4A and Figure 5B).

### 2.7. The Accumulation of ROS

Plant disease resistance relies on the plant’s innate immune system, where ROS and phosphatidic acid (PA) serve as crucial secondary messengers [25]. The direct application of hydrogen peroxide (H_2_O_2_) reagent can inhibit the spore germination of fungi [26]. Additionally, in the pre-invasion stage, the accumulation of H_2_O_2_ occurs in the guard cells of wheat leaves. At 48 hpi, H_2_O_2_ is predominantly generated in the mesophyll cells in contact with the primary haustorial mother cells of *Pst.* With the passage of time, it can be synthesized in the mesophyll cells encircling the necrotic host cells at the infection site. In addition, H_2_O_2_ can be detected within both the necrotic host mesophyll cells and the adjacent mesophyll cells [27] (Figure 6A,B). During this distinct phase of H_2_O_2_ accumulation, the *Pst* is in a developmental stage that involves it entering the mesophyll cells from adjacent haustorial mother cells until the formation of primary haustoria. This finding indicates that the generation of H_2_O_2_ is triggered subsequent to the recognition of elicitors released from the haustoria of the avirulent race by the receptors of the host cells [27].

### 2.8. Metabolic Network Analysis

KEGG is a database for systematic analysis of gene functions and genomic information [28]. It assists researchers in studying genes and their expression information as an integrated network [29]. KEGG can connect a series of genes in the genome through an intracellular molecular interaction network, thereby presenting higher-level biological functions [30]. During the infection of AM1350 by *Pst*, multiple metabolic pathways are involved. The activities of some enzymes in the metabolic pathways of ascorbate and aldarate show a downregulated phenomenon, which is the result of the downregulation of the expression of some genes in AM1350 after being infected by *Pst*. Ascorbic acid (AsA) is an important antioxidant in plants and can scavenge ROS [31,32]. Myo-inositol is one of the starting substances in the metabolic pathways of ascorbic acid and aldarate. It is cleaved into glucuronic acid, which is then converted into AsA via L-gulonate [33,34]. After AM1350 is infected by *Pst*, the gene corresponding to superoxide dismutase (EC 1.13.99.1), the enzyme that catalyzes the synthesis of L-gulonate from myo-inositol, is downregulated. This reduces the consumption of myo-inositol and decreases the production of AsA, thus lowering the AsA content in plant cells. Moreover, monodehydroascorbate is used to synthesize AsA under the action of monodehydroascorbate reductase (EC 1.6.5.4). However, after the infection of *Pst*, the expression level of EC 1.6.5.4 decreases, inhibiting the process of monodehydroascorbate being used to synthesize AsA, which results in a reduction in the accumulation of AsA. Glutathione dehydrogenase (EC 1.8.5.1) catalyzes the production of AsA. Nevertheless, the infection of *Pst* leads to the downregulation of the expression of the differentially expressed genes (DEGs) corresponding to EC 1.8.5.1, and the content of EC 1.8.5.1 in plant cells decreases. Consequently, the synthesis of AsA is reduced (Figure 6C).

## 3. Discussion

### 3.1. Upregulation and Downregulation of DEGs

Genes with various types of functional connections as defined in KEGG pathways often have positively correlated expression levels. Therefore, in wheat cells, genes within the disrupted pathways tend to be upregulated or downregulated similarly due to their close functional associations. It is worth noting that many researchers frequently refer to KEGG pathways enriched with upregulated or downregulated genes as activated or inhibited pathways [35]. However, it may not be appropriate to simply consider a KEGG pathway as an activated or inhibited one merely based on the enrichment of upregulated or downregulated genes [36,37]. During the infection of wheat by *Pst*, numerous DEGs are present, and these DEGs show different degrees of upregulation or downregulation at different time points. These results may suggest that the observed upregulation patterns could be the outcome of promoting an effective plant resistance response by activating multiple resistance genes. This, in turn, induces a coping mechanism on the fungal side in the form of more extensive expression of defense- and pathogenesis-related genes.

### 3.2. The Role of ROS in the Resistance of AM1350 to Pst

The production of ROS during pathogen infection of plants is related to inducer receptors, G-proteins, calcium ions, and protein kinases. Under normal circumstances, the main sites of ROS production in plants are mitochondria and chloroplasts. However, after pathogen infection, a large amount of ROS is also detected outside the cell membrane, which may originate from the NADPH oxidase on the cell membrane [38,39]. The ROS produced in chloroplasts acts as a signaling molecule to transmit information, initiating the defense system in the plant and restricting the infection of *Pst*. When a pathogen infects a plant, the plant rapidly generates a large amount of ROS at the infection site, and then activates other signal cascade systems in the body to resist the pathogen. This process is called oxidative burst. Exogenous H_2_O_2_ in plants has a direct toxic effect on both fungi and bacteria, inhibiting the germination of fungal spores and the growth of bacteria [40,41,42]. In the early stages of *Pst* infecting AM1350, the “Peroxisome” pathway is significantly enriched. The accumulation of ROS increases, enhancing the toxic effect on pathogen [43]. After the pathogen invades plant tissues, necrotic spots form near the infection sites. The uninfected plant tissues around the pathogen spots produce phytoalexins to inhibit the spread of the pathogen. The synthesis of phytoalexins is closely related to the production of ROS [44,45,46]. The production of ROS promotes the synthesis of phytoalexins, inhibiting the spread of the pathogen at the pathogen infected sites.

### 3.3. Analysis of the Reasons for the Changes in Mycelial Length and Area

During the *Pst* infection process, the changes in mycelial length and area are not uniformly linear. Within the range of 24–48 hpi, the changes in the length and area of *Pst* mycelia are relatively small. A reason is that the activity of cell wall degrading enzymes is low at 24 hpi, increases significantly after 48 hpi, peaks at 48 or 72 hpi, and then gradually decreases [47]. The acceleration of the cell wall degradation rate is conducive to the expansion of mycelia and the formation of ultrastructures such as haustoria [48]. This results in relatively small changes in the length and area of mycelia in the early stage, while significant changes occur from 48 to 72 hpi. Meanwhile, according to the elicitor–receptor model theory, specific signal molecules encoded by the avirulence genes of pathogens are recognized by the receptors encoded by the host resistance genes, leading to certain defense responses of plants against pathogen invasion. *Pst* has developed a mechanism to cope with the ROS stress caused by the increase of ROS, secreting effector proteins to relieve the toxicity of ROS, resulting in a significant increase in mycelial area and length. From 72 to 120 hpi, the accumulation of ROS increases, enhancing the toxic effect on *Pst* and slowing down the expansion rate and area of mycelia.

### 3.4. Plant–Pathogen Interactions

The interactions between plants and pathogens are classified into compatible interactions and incompatible interactions. In a compatible interaction, the pathogen can successfully infect the plant, leading to plant disease. In an incompatible interaction, the plant can recognize the pathogen, activate defense responses, and prevent the pathogen from infecting the plant. Plants recognize pathogen-associated molecular patterns (PAMPs) via pattern-recognition receptors (PRRs) on the cell surface, triggering PAMP-triggered immunity (PTI). Additionally, plants can recognize pathogen effectors through resistance proteins, initiating effector-triggered immunity (ETI) [49,50]. Plants can be infected by a variety of pathogens, which manipulate host cells to enable their growth and spread [51]. Pathogens employ diverse manipulation strategies that change during the infection process. These strategies include suppressing plant defenses and promoting colonization and nutrient release using toxins and degrading enzymes [52]. To enrich DEGs in the KEGG pathway of “plant–pathogen interactions”, we found that many DEGs are involved in the PTI and ETI pathways [53]. The expression level of “plant–pathogen interactions” pathway increases significantly in the late stage of *Pst* infection (Appendix A). It has been reported that damage-induced ROS, including H_2_O_2_, provide additional protection against *Botrytis cinerea* in *Arabidopsis thaliana* [54]. The accumulation of this ROS is associated with the permeability of the plant cuticle and the level of abscisic acid (ABA) [55]. The apoplastic generation of H_2_O_2_ has been documented following the recognition of various pathogens [56]. In the early stage of infection (6–24 hpi), ROS, acting as signaling molecules, activate the expression of wheat defense genes, restricting the initial infection of *Pst* [41]. The protective enzyme systems and antioxidant substances in the ROS scavenging system are damaged, further increasing the accumulation of ROS. This exerts a toxic effect on the pathogen, reducing the number of infection sites of the pathogen [42]. At multiple time points (48 hpi, 72 hpi, 120 hpi), plant hormone signal transduction was significantly enriched. The produced plant hormone ABA could enhance the resistance [57]. Meanwhile, the activity of chitinase increased, reducing the pathogen infection [58]. In the early stage of *Pst* infecting AM1350 (6–24 hpi), pathways such as “Peroxisome”, and “Glycine, serine and threonine metabolism” acted as signaling molecules to activate plant immunity [42]. AM1350 continuously activates defense mechanisms, producing more antibacterial substances, strengthening the defensive structure of the cell wall, etc., to resist the further infection of *Pst* [59,60]. At 120 hpi, the “MAPK signaling pathway–plant” and plant hormone signal transduction are significantly enriched. The “MAPK signaling pathway–plant” can activate defense-related transcription factors through phosphorylation, regulate the expression of defense genes, and enhance the resistance of wheat to stripe rust [61,62].

## 4. Materials and Methods

### 4.1. Plant Materials and Pst Inoculation

AM1350 is a semi-winter wheat variety. Its seedlings are semi-prostrate, with dark green leaves, vigorous seedling growth, and strong tillering ability. It starts to joint early in spring, undergoes rapid differentiation between productive and unproductive tillers, and has an early heading date. The variety exhibits good winter hardiness, with 367,000 to 424,000 spikes per mu, 32.2 to 34.6 grains per spike, and a 1000-grain weight of 42.9 to 48.4 g. The average yield is 589.7 kg per mu. It is suitable for sowing in early to mid-October in Henan Province and other regions. Under greenhouse conditions, the wheat variety AM1350 was chosen for this study. Wheat seeds were sown in small pots, with six seedlings planted in each pot and a total of 42 pots used. They were cultivated at 16 °C under a photoperiod of 16 h of light and 8 h of dark until the two-leaf stage. Phenotypic identification was conducted on AM1350 plants that had grown to the stage of fully expanded two-leaf in the incubator. *Pst* race CYR34 was inoculated onto the second leaves. Fresh urediniospores were collected and added to the electronic fluorinated liquid 3M^TM^Novec^TM^7100, then the mixture was thoroughly homogenized. Using a pipette, 10 μL of the mixture was evenly inoculated onto the leaves of wheat. After inoculation, the leaves were evenly sprayed with water to provide the necessary humidity for spore hydration, swelling, and germination to produce germ tubes, thereby improving infection efficiency, kept in the dark and moisturized for 24 h, then placed in a 16 °C light incubator (16 h light/8 h dark cycle). Histological samples and RNA samples were collected at different time points (0 hpi, 6 hpi, 12 hpi, 24 hpi, 48 hpi, 72 hpi, and 120 hpi) for RNA-seq analysis, with the leaf samples collected at 0 hpi serving as the control group. Regulated by the genome, during the process of *Pst* infecting wheat, at 6 hpi, the urediniospores of *Pst* germinated on the wheat leaf surface, producing germ tubes [63]. These germ tubes extended towards the stomata and penetrated into the leaf tissues through the stomata. At 12 hpi, primary hyphae were generated and primary haustorial mother cells were formed. At 24 hpi, the formation of primary haustoria could be observed at most infection sites. At 48 hpi, secondary hyphae and secondary haustoria began to form, and as the pathogen expanded, colonies were formed. We collected samples within one minute after inoculation as 0 hpi samples, designating them as the control group. Other samples were collected at 6 hpi, 12 hpi, 24 hpi, 48 hpi, 72 hpi, and 120 hpi, with each time point having 3 biological replicates [23]. After collection, all leaf samples were promptly plunged into liquid nitrogen, then stored at -80 °C. They remained in this state until RNA extraction was carried out for sequencing. Due to the inability to observe phenotypes on leaves during the early infection stage, we did not display phenotypic images. Meanwhile, we selected to stain the hyphae for observing the developmental stages.

### 4.2. RNA Extraction, Library Construction and Sequencing

The library construction was entrusted to an RNA-seq company (Novogene Co., Ltd.,China). The company’s returned report specified the methodology for library construction, and we briefly introduce the procedures as follows: RNA extraction was performed according to the method provided by the Huayueyang Plant RNA Extraction Kit. The library preparation kit used in library construction is Illumina’s NEBNext^®^ Ultra™ RNA Library Prep Kit. Before RNA extraction, agarose gel electrophoresis was performed to analyze the integrity of sample RNA and the presence of DNA contamination rRNA. No smearing was observed in the bands, indicating the absence of genomic DNA contamination [23]. The NanoPhotometer^®^ NP80 (Implen GmbH, Munich, Germany) was used to detect the purity of RNA, with reference to the ratios of OD260/280 and OD260/230. The total amounts and integrity of RNA were assessed using the RNA Nano 6000 Assay Kit of the Bioanalyzer 2100 system (Agilent Technologies, Santa Clara, CA, USA). The pure RNA ratio is 1.8–2.1. A value significantly higher than 2.1 may indicate DNA contamination. The results show that the OD260/280 and OD260/230 ratios of the RNA used for library construction fall within 1.8–2.1, confirming no DNA contamination. For library construction, total RNA serves as the starting material. Oligo (dT) magnetic beads are used to enrich mRNA with polyA tails. Fragmented mRNA acts as the template, and random oligonucleotides are used as primers to synthesize the first strand of cDNA in the M–MuLV reverse transcriptase system. Subsequently, RNaseH is employed to degrade the RNA strand, and in the DNA polymerase I system, dNTPs are used as raw materials to synthesize the second strand of cDNA. The purified double–stranded cDNA undergoes end-repair, A-tail addition, and sequencing adapter ligation. AMPure XP beads are used to screen cDNA around 370–420 bp. Then, PCR amplification is carried out, and the PCR products are purified again using AMPure XP beads to obtain the final library for sequencing. We collected samples at 0 h, 6 h, 12 h, 24 h, 48 h, 72 h, and 120 h post-inoculation, with three biological replicates for each time point. One gene library was constructed for each sample, resulting in a total of 21 libraries successfully built. After library construction is completed, the Qubit2.0 Fluorometer is first used for preliminary quantification, and the library is diluted to 1.5 ng/ul. Subsequently, the Agilent 2100 bioanalyzer is used to detect the insert size of the library. During the amplification process in the flow cell for sequencing, four fluorescently labeled dNTPs, DNA polymerase, and adapter primers are added. The sequencer detects the fluorescent signals, and the software converts these optical signals into sequencing peaks, thereby obtaining the sequence information of the fragments to be tested [2].

### 4.3. Differential Expression Analyses of Genes

The raw sequencing data presented in the FASTQ format undergoes a filtering process to eliminate reads of low quality, including those with adapters, ambiguous bases, or poor sequencing quality. Clean data is then assessed for Q20, Q30, and GC content to ensure quality. HISAT2 (v2.0.5) is used to align clean reads to the reference genome, chosen for its ability to handle splice junctions effectively. StringTie (1.3.3b) predicts novel transcripts and estimates gene expression more accurately and efficiently. Gene expression levels are calculated using FeatureCounts (1.5.0-p3) and normalized by Fragments Per Kilobase of exon model per Million mapped fragments (FPKM), accounting for sequencing depth and gene length. DESeq2 (1.20.0) is used for differential expression analysis with biological replicates, and edgeR (3.24.3) is used when replicates are unavailable. Genes with padj ≤ 0.05 and |log2 (fold change)| ≥ 1 are considered significantly differentially expressed. Transcriptome analysis, which involves a large number of genes, has the potential to result in the build-up of false positives. As the number of genes analyzed increases, the degree of false positive accumulation in hypothesis testing also rises. To tackle this issue, the padj value is introduced to rectify the *p*-value obtained from hypothesis testing. The Bonferroni correction method is employed to regulate the proportion of false positives [64].

### 4.4. GO Terms and KEGG Enrichment Analyses of DEGs

Gene Ontology (GO) enrichment analysis of DEGs was performed using the clusterProfiler (3.8.1) software, which corrects for gene length bias. GO terms with an adjusted *p*-value of less than 0.05 were considered significantly enriched among the DEGs. Kyoto Encyclopedia of Genes and Genomes (KEGG) pathway enrichment, which offers in-depth understandings of biological functions from the molecular perspective, was also analyzed using clusterProfiler to identify statistically enriched pathways among the DEGs.

### 4.5. Identification and Analysis of Transcription Factors

Weighted Gene Co-expression Network Analysis (WGCNA) is a methodology in systems biology that serves to depict the patterns of correlation existing between genes when analyzed across various samples. It helps identify gene sets that show highly coordinated expression patterns and, based on the connectivity within these gene sets and their association with phenotypes, selects potential biomarker genes or therapeutic targets [65]. The R package of WGCNA offers a diverse array of functions for weighted correlation analysis. These functions span across multiple aspects such as network construction, gene screening, module identification, calculation of topological features, data simulation, and visualization.

### 4.6. Diaminobenzidine Staining and Observation of Samples

The Diaminobenzidine (DAB) staining method was used to observe and count the area of H_2_O_2_ accumulation near the infection sites of AM1350 plants after inoculation with *Pst* race CYR34. The inoculated leaves were collected and placed in a 1 mg/mL DAB solution (pH 3.8) and exposed to strong light for 6 h. Then, the leaves were cut into segments about 2 cm long, and a decolorizing solution (anhydrous ethanol:glacial acetic acid in a volume ratio of 1:1) was added. The samples were decolorized until the leaves became transparent. A saturated chloral hydrate solution was added for fixation for 24 h. After that, the samples were washed three times with a 10% glycerol solution, then placed in a 10% glycerol solution. The stained leaves were taken and placed on a glass slide, and the area of reactive oxygen species (ROS) was observed and counted using a microscope.

### 4.7. Quantitative Real-Time PCR

Four DEGs were randomly selected from the pathways significantly enriched following *Pst* infection in wheat. Gene-specific primers were designed using Primer Premier 5.0, targeting conserved regions of the selected DEGs. qRT-PCR was performed on Bio-Rad CFX connect Real-Time PCR System, according to the manufacturer’s instructions with cDNA templates derived from wheat cultivar Fielder inoculated with *Pst* race CYR34 at multiple time points post-inoculation. The PCR reaction mix (20 μL total volume) contained 1 μL of diluted cDNA template (1:10 dilution), 10 μL of 2× ChamQ Blue Universal SYBR qRT-PCR Master Mix, and 0.4 μL of each primer (10 μM final concentration). Reactions were performed under standard conditions: initial denaturation at 95 °C for 30 s, followed by 40 cycles of 95 °C for 10 s and 60 °C for 30 s, with melt curve analysis from 65 °C to 95 °C. Relative gene expression levels were calculated using the 2^−ΔΔCt^ method, normalized against the endogenous reference gene *TaEF-1α* (wheat elongation factor 1-α, Gene ID: TraesCS5D02G423400). *TaEF-1α* belongs to housekeeping genes, and its encoded protein is involved in the elongation process of intracellular protein synthesis, which is essential for basic cellular life activities. The mRNA expression level of *TaEF-1α* is generally relatively constant across different tissues, developmental stages, or treatment conditions [21]. Each biological replicate (n = 3) consisted of independently extracted RNA samples, with three technical replicates per biological sample. The list of qRT-PCR primer sequences can be found in Appendix A.

## 5. Conclusions

Analyzing the transcriptomic changes in AM1350 at different time points is the first step towards understanding the molecular mechanism of AM1350’s resistance to *Pst*. The transcriptomic data generated here will contribute to guiding advanced research for developing efficient methods for disease-resistant wheat production. By examining the changes in ROS accumulation within AM1350 at different stages and the relationships among different metabolic pathways, we have clarified that AM1350 exhibits remarkable resistance to *Pst*. In the early stage, AM1350 accumulates energy through metabolic pathways such as glycolysis, which provides a material basis for immune response. In the later stage, the increased accumulation of ROS promotes the synthesis of phytoalexins, which are toxic to *Pst* and inhibit its infection, enabling AM1350 to display resistance to *Pst*. Meanwhile, the pathways involved in the “plant–pathogen interaction” process have been studied, offering new perspectives for researching how to enhance plant resistance.

## Figures and Tables

**Figure 1 ijms-26-05538-f001:**
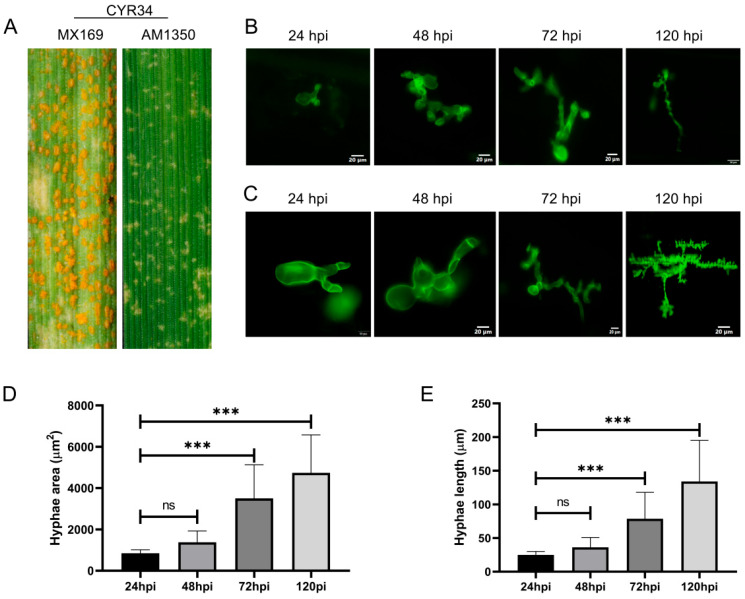
Phenotypes of leaves of MX169 and AM1350 after inoculation with *Pst* race CYR34 respectively (**A**). Mycelium images at different stages stained with WGA in AM1350 (**B**). Mycelium images at different stages stained with WGA in AM1350 (**C**). Mycelial lengths within AM1350 cells at different stages. Asterisks indicate a significant difference (ns means “non - significant”, and *** means *p* < 0.001) (**D**). Mycelial areas within AM1350 cells at different stages. Asterisks indicate a significant difference (ns means “non—significant”, *** means *p* < 0.001) (**E**).

**Figure 2 ijms-26-05538-f002:**
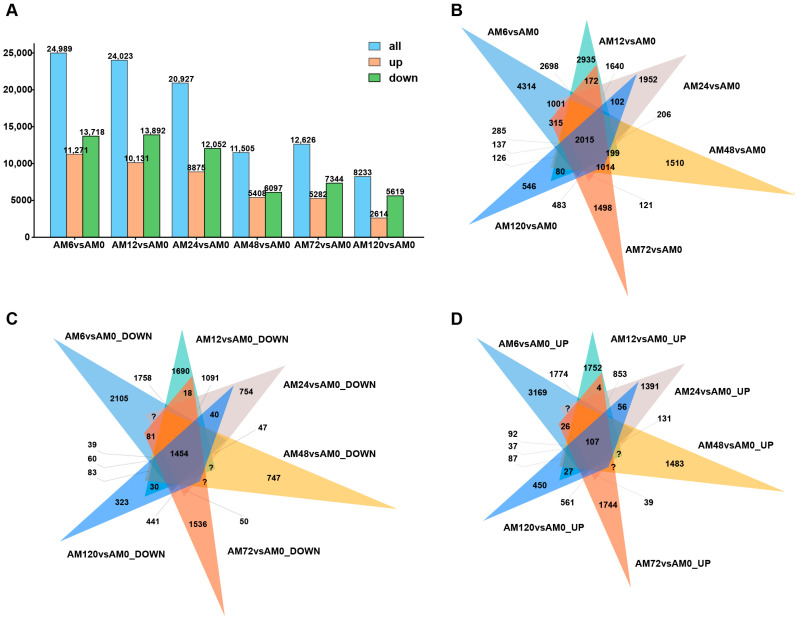
Number of DEGs, as well as the number of upregulated and downregulated DEGs in AM1350 at 6 hpi, 24 hpi, 48 hpi, 72 hpi, and 120 hpi (**A**). Expression levels of DEGs in AM1350 at different time points in a Venn diagram (**B**). Expression levels of downregulated part of DEGs in AM1350 at different time points in a Venn diagram (**C**). Expression levels of upregulated part of DEGs in AM1350 at different time points in a Venn diagram (**D**).

**Figure 3 ijms-26-05538-f003:**
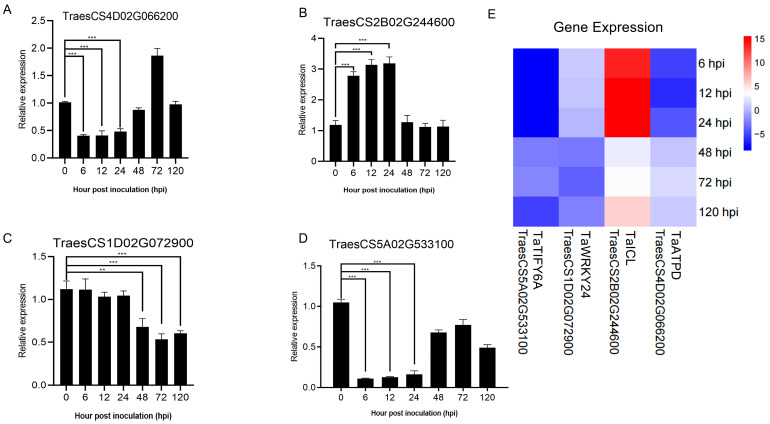
Quantitative analysis of four DEGs by qRT-PCR using cDNA templates from wheat cultivar Fielder inoculated with *Pst* race CYR34 at different time points. Asterisks indicate a significant difference (** means *p* < 0.01, *** means *p* < 0.001) (**A**–**D**). Heat map of RNA-seq data analysis for AM1350 at different time points after inoculation with *Pst* (**E**).

**Figure 4 ijms-26-05538-f004:**
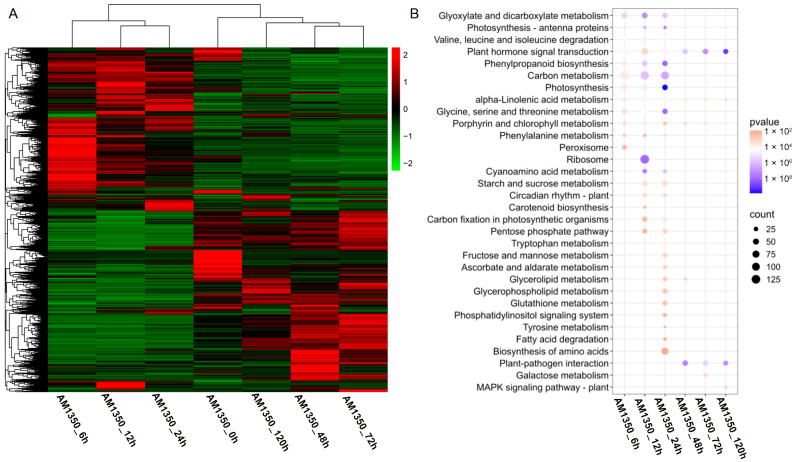
Cluster heatmap of DEGs (**A**). KEGG enrichment analysis of DEGs; only significantly enriched pathways were showed (**B**).

**Figure 5 ijms-26-05538-f005:**
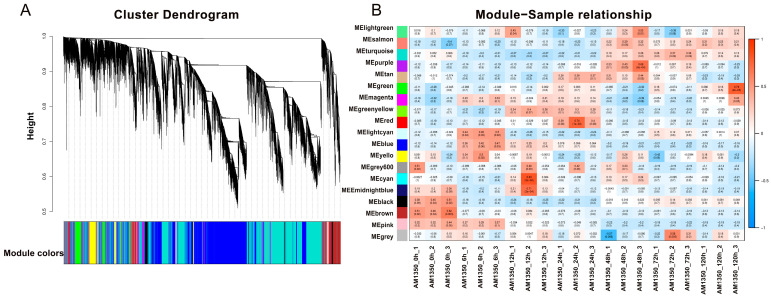
Modular hierarchical clustering dendrogram of correlations in gene expression levels among genes (**A**); heatmap of correlations among modules (**B**).

**Figure 6 ijms-26-05538-f006:**
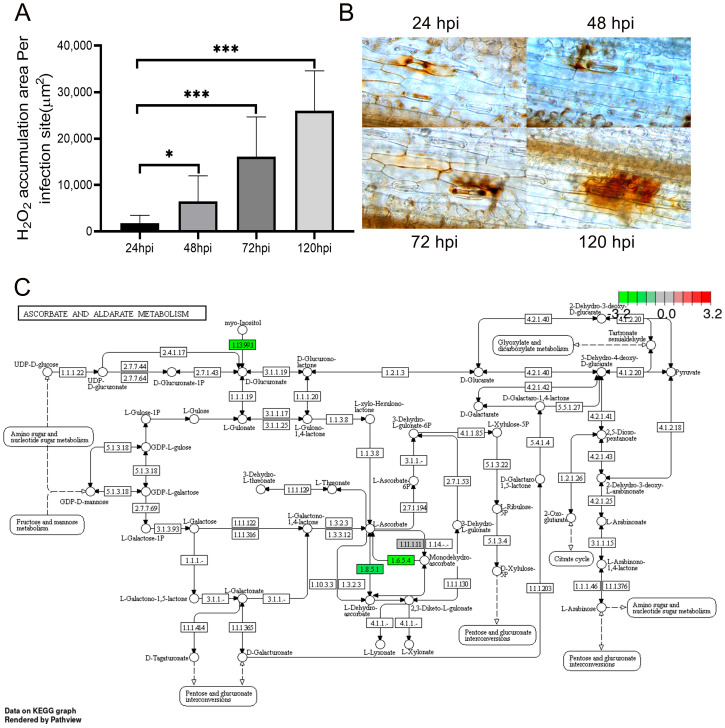
Area of H_2_O_2_ accumulation region at each infection site. Asterisks indicate a significant difference (* means *p* < 0.05, *** means *p* < 0.001) (**A**). Staining of H_2_O_2_ accumulation region in guard cells at 24 hpi, 48 hpi, 72 hpi, and 120 hpi after *Pst* infection of AM1350 (**B**). Changes in gene expression levels of “ASCORBATE AND ALDARATE METABOLISM” pathway during infection of AM1350 by *Pst* (**C**).

**Table 1 ijms-26-05538-t001:** Mapping high-quality reads to the reference genome.

Sample	Clean_Bases	Total_Reads	Error_Rate (%)	Q20 (%)	Q30 (%)	GC_pct (%)	Total_Map	Mapping Rate (%)
AM1350_0h_1	6.94G	46,272,540	0.03	97.61	93.42	55.37	43,222,298	93.41
AM1350_0h_2	6.92G	46,119,588	0.03	97.54	93.22	56.27	43,144,542	93.55
AM1350_0h_3	6.68G	44,541,752	0.03	97.68	93.57	53.80	41,576,571	93.34
AM1350_6h_1	6.93G	46,229,954	0.03	97.48	93.13	55.80	42,768,413	92.51
AM1350_6h_2	6.73G	44,852,278	0.03	97.56	93.29	55.82	41,577,526	92.70
AM1350_6h_3	6.89G	45,964,126	0.03	97.56	93.27	54.73	42,404,143	92.25
AM1350_12h_1	6.79G	45,272,470	0.03	97.55	93.26	54.45	41,741,226	92.20
AM1350_12h_2	6.36G	42,380,874	0.03	97.48	93.16	54.81	39,218,703	92.54
AM1350_12h_3	6.18G	41,192,308	0.03	97.54	93.30	54.97	38,155,268	92.63
AM1350_24h_1	6.19G	41,263,874	0.03	97.49	93.23	54.52	37,915,463	91.89
AM1350_24h_2	7.13G	47,543,002	0.03	97.64	93.53	55.62	43,969,645	92.48
AM1350_24h_3	7.44G	49,590,532	0.03	97.22	92.49	54.68	45,738,645	92.23
AM1350_48h_1	7.24G	48,254,104	0.03	97.61	93.43	54.48	44,718,819	92.67
AM1350_48h_2	6.63G	44,228,828	0.03	97.46	93.08	55.82	40,993,286	92.68
AM1350_48h_3	6.56G	43,727,762	0.03	97.78	93.73	56.35	40,682,044	93.03
AM1350_72h_1	6.46G	43,076,234	0.03	97.56	93.25	53.00	38,587,764	89.58
AM1350_72h_2	6.18G	41,231,804	0.03	97.74	93.69	55.43	38,759,419	94.00
AM1350_72h_3	6.43G	42,840,026	0.03	97.70	93.63	55.47	40,041,500	93.47
AM1350_120h_1	6.27G	41,795,472	0.03	97.40	92.94	55.82	38,686,028	92.56
AM1350_120h_2	6.35G	42,331,432	0.03	97.65	93.49	55.76	39,374,360	93.01
AM1350_120h_3	6.89G	45,923,630	0.03	97.31	92.80	53.31	40,694,489	88.61

## Data Availability

The datasets presented in this study are publicly available. This data can be found in the SRA database in NCBI (link: https://www.ncbi.nlm.nih.gov/bioproject/PRJNA1265958, accessed on 4 June 2025; accession: PRJNA1265958).

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
