# Peer review of "Transcriptome Analysis Reveals Mechanisms of Stripe Rust Response in Wheat Cultivar Anmai1350"

_ijms, 2025, doi:10.3390/ijms26125538_

Round 1
Reviewer 1 Report
Comments and Suggestions for Authors
page 11 line 347 pls check the reference in this sentence.
The original sequencing data ( raw data or clean data) must be submitted to a public database( e.g. CHINA national genebank) , All the I.D. and their relevant sequences generated in the RNA-seq sequencing experiments as well as the original sequencing counts (expression level) of all genes in the 21 samples of this study must also be disclosed, otherwise this paper is meaningless to readers without these scientific information. I am saying that without the help with these information, the readers can not know which I.D refer to which gene , and what is its sequence and also what is its expression level.
Author Response
Comment 12: The original sequencing data ( raw data or clean data) must be submitted to a public database( e.g. CHINA national genebank) , All the I.D. and their relevant sequences generated in the RNA-seq sequencing experiments as well as the original sequencing counts (expression level) of all genes in the 21 samples of this study must also be disclosed, otherwise this paper is meaningless to readers without these scientific information. I am saying that without the help with these information, the readers can not know which I.D refer to which gene , and what is its sequence and also what is its expression level.
Response 12: Regarding our raw data, we are in the process of uploading it to NCBI. However, due to the information settings on the website, our excessively large data compression package cannot be uploaded to the NCBI database for the time being. There are a total of 42 original data sets. We have currently uploaded 25 of them, and the subsequent upload is expected to be completed within three days.
Reviewer 2 Report
Comments and Suggestions for Authors
Please check the attached file.

Author Response
Comment 1: Figure 3B looks interesting to me. Point 1: Genes related to “Photosynthesis” and “Carbon metabolism” were differentially expressed at each time point at Day 1. In contrast, these genes were not differentially expressed on day 2 and n3. Point 2: While genes related to “Plant hormone signal transduction” and “Plant-pathogen interaction” were differentially expressed on day 2 and 3. These two points can direct a better understanding of wheat in response to Pst.
Response 1:We fully agree with the reviewer's suggestion and have made the corresponding modifications in the manuscript. We conducted a more comprehensive investigation of the significantly enriched pathways in Figure 3B, explaining the roles of processes such as photosynthesis and carbon metabolism in AM1350. This provides a better foundation for subsequent research on the resistance mechanisms of AM1350.
Comment 2: However, I am not sure about the data as the control in the experiment is not clear to me. There is a chance of changes in DEGs and network analysis results once control (samples without Pst inoculation). Line 87: What did you use for the control? Line 95: Does 0 hpi mean just after inoculation?
Response 2: In this paper, the control group was set as the leaf samples of AM1350 collected immediately after inoculation with Pst, and all subsequent analyses of differentially expressed genes (DEGs) were based on this criterion.
Comment 3: Another problem in this manuscript is the lack of RNA-seq data validation. Line 160: In the methodology sections, validation of the RNA-seq data is missing.
Response 3: Thank you for the suggestion. The updated content now reads: The team randomly selected four DEGs to validate the RNA sequencing data through qRT-PCR experiments, demonstrating the authenticity and reliability of the RNA sequencing results(Figure 1).
Figure 1. Quantitative analysis of four DEGs by qRT-PCR using cDNA templates from wheat cultivar Fielder inoculated with Pst race CYR34 at different time points (A-D).
Comment 4: Do not use synonyms suggested by software like Microsoft or others. Otherwise, revise it by a native speaker in plant science
Response 4: Regarding the grammatical errors in the text, we have already made corrections. If there are any inadequacies, please do not hesitate to provide your valuable feedback.
Comment 5: Lines 41-43: I guess it has been copied from https://doi.org/10.1016/j.heliyon.2022.e10951 and paraphrased. Linse 46-50: I guess it has been copied from https://doi.org/10.1016/j.heliyon.2022.e10951 and paraphrased. Lines 46-53: This is a very general statement. Please reduce this portion to 1-2 lines
Response 5: Thank you for the suggestion. The revisions are as follows: Additionally, we provided a more scientific explanation for the analysis of the prevalence of CYR34 in the introduction, noting that 82% of the varieties are susceptible to one or more races of Pst (CYR32, CYR33, and CYR34). Stripe rust mainly overwinters in overwintering regions, where wheat materials are vulnerable to the Pst race CYR34.
Comment 6: Please mention properly about AM1350.Lines 75-76: It is difficult to understand “early and medium sown fields”
Response 6: We fully agree with your comments and have made the following revisions: AM1350 is a semi-winter wheat variety. Its seedlings are semi-prostrate, with dark green leaves, vigorous seedling growth, and strong tillering ability. It starts jointing early in spring, undergoes rapid differentiation between productive and unproductive tillers, and has an early heading date. The variety exhibits good winter hardiness, with 367,000 to 424,000 spikes per mu, 32.2 to 34.6 grains per spike, and a 1000-grain weight of 42.9 to 48.4 grams. The average yield is 589.7 kilograms per mu. It is suitable for sowing in early to mid-October in Henan Province and other regions.
Comment 7: Lines 54-55: “However, a comprehensive gene expression profile of wheat in response to Pst remains unavailable” ‒ this statement is not correct. https://doi.org/10.1128/spectrum.03774-23, https://doi.org/10.1016/j.heliyon.2022.e10951
Response 7: The revision is as follow: However,a complete gene expression profile of wheat in response to stripe rust pathogens is still lacking.
Comment 8: Line 78: with six seedlings planted in each pot?? Or six seeds sown in each pot? How many pots were used for each treatment? Lines 84-85: What was the purpose of the water spray after inoculation?
Response 8: We highly appreciate your valuable comments and have implemented the following modifications: Wheat seeds were sown in small pots, with six seedlings planted in each pot and a total of 42 pots used. After inoculation, the leaves were evenly sprayed with water to provide the necessary humidity for spore hydration, swelling, and germination to produce germ tubes, thereby improving infection efficiency. The pots were kept in the dark and moisturized for 24 hours, then placed in a 16°C light incubator (16 h light/8 h dark cycle).
Comment 9: Lines 101-102: What was the method for RNA extraction? Which kit did you use? How did you check DNA contamination before RNA extraction? Line 108: How many libraries did you prepare for RNA-seq? Which Kit was used for library preparation?
Response 9: Thank you for your comprehensive modification suggestions. The revisions are as follows: Elaborate explanations were given for the RNA extraction method and library construction used in the experiment. RNA extraction was performed according to the method provided by the Huayueyang Plant RNA Extraction Kit. Before RNA extraction, agarose gel electrophoresis was conducted to analyze the integrity of the sample RNA and the presence of DNA contamination²⁴. The purity of RNA was detected using the NanoPhotometer® NP80 from Implen GmbH (Germany), with reference to the ratios of OD260/280 and OD260/230. After RNA synthesis and purification, double-stranded cDNA was subjected to end repair, A-tailing, and sequencing adaptor ligation. cDNA fragments of approximately 370–420 bp were selected using AMPure XP beads, followed by PCR amplification and re-purification of PCR products using AMPure XP beads, ultimately obtaining the library for sequencing.
Comment 10: Line 197: What are the different stages? Did you mean different hpi?
Response 10: Thank you for your reminder. In the original text, the "different stages" mentioned in Line 197 refer to the time points of 6 hpi, 12 hpi, 24 hpi, 48 hpi, 72 hpi, and 120 hpi after infection.
Comment 11: What is Yr, abbriviate plesae.
Response 11: Thank you for your reminder. We have revised it to Yr26 (Yellow rust resistance 26)

Round 2
Reviewer 2 Report
Comments and Suggestions for Authors
Attached

Author Response
Comment 1: Figure 3B looks interesting to me. Point 1: Genes related to “Photosynthesis” and “Carbon metabolism” were differentially expressed at each time point at Day 1. In contrast, these genes were not differentially expressed on day 2 and n3. Point 2: While genes related to “Plant hormone signal transduction” and “Plant-pathogen interaction” were differentially expressed on day 2 and 3. These two points can direct a better understanding of wheat in response to Pst.
Re-comment 1: I checked your texts in the current version between Lines 283-290. It is not the investigations, it is simply texting what I questioned. Identifying genes from these GO responses to defence mechanisms, and checking their expression by qPCR, could be a way to represent some candidates, and these candidates can be verified by the overexpressed lines.
Response 1: I apologize for not fully understanding your comment during the first revision. Our team has now completed the modifications. The revised content is as follows:
TaATPD (ATP synthase delta chain, Gene ID: TraesCS4D02G066200) related to “Photosynthesis” pathway exhibited significant differential expression at 6 hpi, 12 hpi and 24 hpi. The results of qRT-PCR showed that the expression level exhibited a clear downregulation at 6 hpi, 12 hpi, and 24 hpi. However, at 48 hpi, 72 hpi, and 120 hpi, the expression level showed no differences in gene expression at 0 hpi (Figure 1 A). TaICL (Isocitrate lyase, Gene ID: TraesCS2B02G244600) related to “Carbon metabolism” pathway exhibited significant differential expression at 6 hpi, 12 hpi and 24 hpi. The results of qRT-PCR showed that it had upregulated at 6 hpi, 12 hpi, and 24 hpi. At 48 hpi, gene expression levels decreased (Figure 1 B). TaWRKY24 (WRKY transcription factor WRKY24, Gene ID: TraesCS1D02G072900) on the “Plant-pathogen interaction” pathway exhibited significant differential expression at 48 hpi, 72 hpi, and 120 hpi. The result of qRT-PCR showed downregulation at 24 hpi, 48 hpi, 72 hpi, and 120 hpi (Figure 1 C). TaTIFY6A (TIFY domain-containing protein 6a, Gene ID: TraesCS5A02G533100) related to “Plant hormone signal transduction” pathway showed significant differential expression at 6 hpi, 12 hpi, and 24 hpi. There was an upregulation at 48 hpi, 72 hpi, and 120 hpi compared to that of 6 hpi, 12 hpi, and 24 hpi (Figure 1 D). The detailed qRT-PCR expression levels along with RNA-seq data are presented (Supplemental Table 2). The detailed KEGG enrichment pathways is presented (Supplemental Table 3).
Comment 2: However, I am not sure about the data as the control in the experiment is not clear to me. There is a chance of changes in DEGs and network analysis results once control (samples without Pst inoculation). Line 87: What did you use for the control? Line 95: Does 0 hpi mean just after inoculation?
Re-comment 2: Your answer does not clarify my question, however, I have gone through your text again (section 2.1). It means you have inoculated Pst in all plants → sprayed with water → collected samples at 0 hpi. This 0 hpi is control, right? This is a big issue, which can be a methodological error, especially for expression profiling. Usually, researchers never inoculate with pathogens for control, researchers use mock as a control. Does 0 hr mean 0 minutes? Counting proper time after inoculations is sensitive for gene expression profiling.
Response 2: I apologize for not directly explaining our inoculation details during the first revision. The comprehensive revised content is as follows:
Before conducting the experiment, we reviewed multiple references [1] (Corresponding to the references 23 in the original text) and based on the above references, we collected samples within one minute after inoculation as 0 hpi samples, designating them as the control group. The relevant description has been supplemented and revised in the original text: Line 105-Line 107.
Comment 3: Another problem in this manuscript is the lack of RNA-seq data validation. Line 160: In the methodology sections, validation of the RNA-seq data is missing.
Re-comment 3: Lines 111: How did you check DNA contamination before RNA extraction? Lines 116-133: How many sequence libraries were prepared for sequencing (previously commented in Comment 9)? Lines 188-190: Which machine was used for qPCR? What is relative to TaEF-1α in Figure 1? Where are your primer list sequences for qPCR? Besides, validation of RNA-seq data with qPCR is not like Figure 6 in the manuscript. These figures have only qPCR data; RNA-seq data is missing.
Response 3: Thank you for raising these questions. We will respond to each of them individually as follows:
1, The library construction was entrusted to an RNA-seq company (Novogene Co., Ltd.). The company's returned report specified the methodology for library construction, and we briefly introduce the procedures as follows:
Before RNA extraction, agarose gel electrophoresis was performed to analyze the integrity of sample RNA and the presence of DNA contamination rRNA. No smearing was observed in the bands, indicating the absence of genomic DNA contamination. The NanoPhotometer® NP80 (Implen GmbH, Germany) was used to detect the purity of RNA, with reference to the ratios of OD260/280 and OD260/230. The total amounts and integrity of RNA were assessed using the RNA Nano 6000 Assay Kit of the Bioanalyzer 2,100 system (Agilent Technologies, CA, USA). The pure RNA ratio is 1.8-2.1. A value significantly higher than 2.1 may indicate DNA contamination. The results show that the OD260/280 and OD260/230 ratios of the RNA used for library construction fall within 1.8-2.1, confirming no DNA contamination.
2, We collected samples at 0 hpi, 6 hpi, 12 hpi, 24 hpi, 48 hpi, 72 hpi, and 120 hpi, with three biological replicates for each time point. One gene library was constructed for each sample, resulting in a total of 21 libraries built.
3, The qRT-PCR was performed on Bio-Rad CFX connect Real-Time PCR System, according to the manufacturer’s instructions.
4, TaEF-1α (Gene ID: TraesCS5D02G423400) belongs to housekeeping genes, and its encoded protein is involved in the elongation process of intracellular protein synthesis, which is essential for basic cellular life activities. The mRNA expression level of TaEF-1α is generally relatively constant across different tissues, developmental stages, or treatment conditions [1] (Corresponding to the references 28 in the original text).
5, We have supplemented the primer sequences used in qRT-PCR. The list of qRT-PCR primer sequences can be found (Supplemental Table 4).
6, We have reselected the genes involved in qRT-PCR and generated a heatmap of their corresponding RNA-seq data for comparison (Figure 1).
Comment 4: Lines 54-55: “However, a comprehensive gene expression profile of wheat in response to Pst remains unavailable ” ‒ this statement is not correct. https://doi.org/10.1128/spectrum.03774-23, https://doi.org/10.1016/j.heliyon.2022.e10951
Re-comment 4: The current sentence has remained the same as the previous.
Response 4: We apologize for the misinterpretation of your comments during the first revision.
In the original text, we stated that “However, a comprehensive gene expression profile of wheat in response to Pst remains unavailable”. After being reminded, we realized that the original expression was too one-sided and failed to fully account for the research status of gene expression profiles in wheat. Through careful study of the literature you provided, we have made the following modifications:
However, the gene expression profile of the special disease-resistant cultivar AM1350 in response to the prevalent race CYR34 of Pst remains unclear.
Comment 5: Line 197: What are the different stages? Did you mean a different hpi?
Re-comment 5: Please replace “different stages” with “different hpi”.
Response 5: Thank you for your correction. We have replace “different stages” to “different hpi”.
Additional comments: Why did you not show a comparative leaf phenotype picture at different hpi in AM1350.
Response 6: Leaves in the early stages of infection were selected as samples. At 6 hpi, the uredospores of stripe rust fungus germinated on the wheat leaf surface to produce germ tubes. At 12 hpi, primary hyphae were generated and primary haustorial mother cells were formed. At 24 hpi, the formation of primary haustoria could be observed at most infection sites and hypersensitive necrosis was observed in mesophyll cells at individual infection sites. At 48 hpi, secondary hyphae and secondary haustoria began to develop and the proportion of infection sites with hypersensitive necrosis increased. Due to the inability to observe phenotypes on leaves during the early infection stage, we did not display phenotypic images. Meanwhile, we selected to stain the hyphae for observing the developmental stages.
Additional comments: There are “Error! Reference source not found” in the entire manuscript. Please fix this issue.
Response 7: Thank you for your reminder. The issue of “Error! Reference source not found” in the figure captions is caused by the manuscript. All issues with incorrect citations have been revised.
Reference:
- Ren, Jing, Liang Chen, Jian Liu, Bailing Zhou, Yujie Sha, Guodong Hu, and Junhua Peng. Transcriptomic Insights into the Molecular Mechanism for Response of Wild Emmer Wheat to Stripe Rust Fungus. Frontiers in Plant Science 14 2024: 1320976.
- Zhang, Rui, Zihao Liu, Shijia Zhao, Xiaojing Zhao, Shuaiwu Wang, Xue Li, Deli Lin, et al. TaEF1a Is Involved in Low Phosphorus Stress Responses and Affects Root Development. Plant Growth Regulation 102, no. 1 2024: 227-36.

Round 3
Reviewer 2 Report
Comments and Suggestions for Authors
Attached
or
Comment 1: Figure 3B looks interesting to me. Point 1: Genes related to “Photosynthesis” and “Carbon metabolism” were differentially expressed at each time point at Day 1. In contrast, these genes were not differentially expressed on day 2 and n3. Point 2: While genes related to “Plant hormone signal transduction” and “Plant-pathogen interaction” were differentially expressed on day 2 and 3. These two points can direct a better understanding of wheat in response to Pst.
Re-comment 1: I checked your texts in the current version between Lines 283-290. It is not the investigations, it is simply texting what I questioned. Identifying genes from these GO responses to defence mechanisms, and checking their expression by qPCR, could be a way to represent some candidates, and these candidates can be verified by the overexpressed lines.
Response 1: I apologize for not fully understanding your comment during the first revision. Our team has now completed the modifications. The revised content is as follows:
TaATPD (ATP synthase delta chain, Gene ID: TraesCS4D02G066200) related to “Photosynthesis” pathway exhibited significant differential expression at 6 hpi, 12 hpi and 24 hpi. The results of qRT-PCR showed that the expression level exhibited a clear downregulation at 6 hpi, 12 hpi, and 24 hpi. However, at 48 hpi, 72 hpi, and 120 hpi, the expression level showed no differences in gene expression at 0 hpi (Figure 1 A). TaICL (Isocitrate lyase, Gene ID: TraesCS2B02G244600) related to “Carbon metabolism” pathway exhibited significant differential expression at 6 hpi, 12 hpi and 24 hpi. The results of qRT-PCR showed that it had upregulated at 6 hpi, 12 hpi, and 24 hpi. At 48 hpi, gene expression levels decreased (Figure 1 B). TaWRKY24 (WRKY transcription factor WRKY24, Gene ID: TraesCS1D02G072900) on the “Plant-pathogen interaction” pathway exhibited significant differential expression at 48 hpi, 72 hpi, and 120 hpi. The result of qRT-PCR showed downregulation at 24 hpi, 48 hpi, 72 hpi, and 120 hpi (Figure 1 C). TaTIFY6A (TIFY domain-containing protein 6a, Gene ID: TraesCS5A02G533100) related to “Plant hormone signal transduction” pathway showed significant differential expression at 6 hpi, 12 hpi, and 24 hpi. There was an upregulation at 48 hpi, 72 hpi, and 120 hpi compared to that of 6 hpi, 12 hpi, and 24 hpi (Figure 1 D). The detailed qRT-PCR expression levels along with RNA-seq data are presented (Supplemental Table 2). The detailed KEGG enrichment pathways is presented (Supplemental Table 3).
Re-comment 1-1: I guess, you are discussing about Figure 3. Unfortunately, I do not find any supplementary files. However, the data presentation looks standard now. Please revise the Figure 3 title and describe each properly.
Comment 2: However, I am not sure about the data as the control in the experiment is not clear to me. There is a chance of changes in DEGs and network analysis results once control (samples without Pst inoculation). Line 87: What did you use for the control? Line 95: Does 0 hpi mean just after inoculation?
Re-comment 2: Your answer does not clarify my question, however, I have gone through your text again (section 2.1). It means you have inoculated Pst in all plants → sprayed with water → collected samples at 0 hpi. This 0 hpi is control, right? This is a big issue, which can be a methodological error, especially for expression profiling. Usually, researchers never inoculate with pathogens for control, researchers use mock as a control. Does 0 hr mean 0 minutes? Counting proper time after inoculations is sensitive for gene expression profiling.
Response 2: I apologize for not directly explaining our inoculation details during the first revision. The comprehensive revised content is as follows:
Before conducting the experiment, we reviewed multiple references [1] (Corresponding to the references 23 in the original text) and based on the above references, we collected samples within one minute after inoculation as 0 hpi samples, designating them as the control group. The relevant description has been supplemented and revised in the original text: Line 105-Line 107.
Re-comment 2-1: It would be nice to add pictures of leaf samples after inoculation at each time points (if you have any).
Comment 3: Another problem in this manuscript is the lack of RNA-seq data validation. Line 160: In the methodology sections, validation of the RNA-seq data is missing.
Re-comment 3: Lines 111: How did you check DNA contamination before RNA extraction? Lines 116-133: How many sequence libraries were prepared for sequencing (previously commented in Comment 9)? Lines 188-190: Which machine was used for qPCR? What is relative to TaEF-1α in Figure 1? Where are your primer list sequences for qPCR? Besides, validation of RNA-seq data with qPCR is not like Figure 6 in the manuscript. These figures have only qPCR data; RNA-seq data is missing.
Response 3: Thank you for raising these questions. We will respond to each of them individually as follows:
1, The library construction was entrusted to an RNA-seq company (Novogene Co., Ltd.). The company's returned report specified the methodology for library construction, and we briefly introduce the procedures as follows:
Before RNA extraction, agarose gel electrophoresis was performed to analyze the integrity of sample RNA and the presence of DNA contamination rRNA. No smearing was observed in the bands, indicating the absence of genomic DNA contamination. The NanoPhotometer® NP80 (Implen GmbH, Germany) was used to detect the purity of RNA, with reference to the ratios of OD260/280 and OD260/230. The total amounts and integrity of RNA were assessed using the RNA Nano 6000 Assay Kit of the Bioanalyzer 2,100 system (Agilent Technologies, CA, USA). The pure RNA ratio is 1.8-2.1. A value significantly higher than 2.1 may indicate DNA contamination. The results show that the OD260/280 and OD260/230 ratios of the RNA used for library construction fall within 1.8-2.1, confirming no DNA contamination.
2, We collected samples at 0 hpi, 6 hpi, 12 hpi, 24 hpi, 48 hpi, 72 hpi, and 120 hpi, with three biological replicates for each time point. One gene library was constructed for each sample, resulting in a total of 21 libraries built.
3, The qRT-PCR was performed on Bio-Rad CFX connect Real-Time PCR System, according to the manufacturer’s instructions.
4, TaEF-1α (Gene ID: TraesCS5D02G423400) belongs to housekeeping genes, and its encoded protein is involved in the elongation process of intracellular protein synthesis, which is essential for basic cellular life activities. The mRNA expression level of TaEF-1α is generally relatively constant across different tissues, developmental stages, or treatment conditions [1] (Corresponding to the references 28 in the original text).
5, We have supplemented the primer sequences used in qRT-PCR. The list of qRT-PCR primer sequences can be found (Supplemental Table 4).
6, We have reselected the genes involved in qRT-PCR and generated a heatmap of their corresponding RNA-seq data for comparison (Figure 1).
Re-comment 3-1: Please upload the supplementary (I think it is missing).
Comment 4: Lines 54-55: “However, a comprehensive gene expression profile of wheat in response to Pst remains unavailable ” ‒ this statement is not correct. https://doi.org/10.1128/spectrum.03774-23, https://doi.org/10.1016/j.heliyon.2022.e10951
Re-comment 4: The current sentence has remained the same as the previous.
Response 4: We apologize for the misinterpretation of your comments during the first revision.
In the original text, we stated that “However, a comprehensive gene expression profile of wheat in response to Pst remains unavailable”. After being reminded, we realized that the original expression was too one-sided and failed to fully account for the research status of gene expression profiles in wheat. Through careful study of the literature you provided, we have made the following modifications:
However, the gene expression profile of the special disease-resistant cultivar AM1350 in response to the prevalent race CYR34 of Pst remains unclear.
Re-comment 4-1: It sounds better. But I would suggest you as to write as followings
The molecular mechanisms underlying the interaction between wheat and Pst can be deciphered through global gene expression profiling approaches such as RNA sequencing
(RNA-seq). RNA-seq has been widely applied to investigate the mechanisms involved in
the interaction between wheat and pathogenic fungi [10,11,12,13,14]. However, resistance mechanisms can be race-specific and can be dependent on the resistance mechanisms of host plant. Despite of wide ranges of molecular studies different host plants, the resistance mechanisms of the AM1350 against CYR34 of Pst is unclear. However, the gene expression profile of the special disease-resistant cultivar AM1350 in response to the prevalent race CYR34 of Pst remains unclear [9].

Author Response
Comment 1: Figure 3B looks interesting to me. Point 1: Genes related to “Photosynthesis” and “Carbon metabolism” were differentially expressed at each time point at Day 1. In contrast, these genes were not differentially expressed on day 2 and n3. Point 2: While genes related to “Plant hormone signal transduction” and “Plant-pathogen interaction” were differentially expressed on day 2 and 3. These two points can direct a better understanding of wheat in response to Pst.
Re-comment 1: I checked your texts in the current version between Lines 283-290. It is not the investigations, it is simply texting what I questioned. Identifying genes from these GO responses to defence mechanisms, and checking their expression by qPCR, could be a way to represent some candidates, and these candidates can be verified by the overexpressed lines.
Response 1: I apologize for not fully understanding your comment during the first revision. Our team has now completed the modifications. The revised content is as follows:
TaATPD (ATP synthase delta chain, Gene ID: TraesCS4D02G066200) related to “Photosynthesis” pathway exhibited significant differential expression at 6 hpi, 12 hpi and 24 hpi. The results of qRT-PCR showed that the expression level exhibited a clear downregulation at 6 hpi, 12 hpi, and 24 hpi. However, at 48 hpi, 72 hpi, and 120 hpi, the expression level showed no differences in gene expression at 0 hpi (Figure 3 A). TaICL (Isocitrate lyase, Gene ID: TraesCS2B02G244600) related to “Carbon metabolism” pathway exhibited significant differential expression at 6 hpi, 12 hpi and 24 hpi. The results of qRT-PCR showed that it had upregulated at 6 hpi, 12 hpi, and 24 hpi. At 48 hpi, gene expression levels decreased (Figure 3 B). TaWRKY24 (WRKY transcription factor WRKY24, Gene ID: TraesCS1D02G072900) on the “Plant-pathogen interaction” pathway exhibited significant differential expression at 48 hpi, 72 hpi, and 120 hpi. The result of qRT-PCR showed downregulation at 24 hpi, 48 hpi, 72 hpi, and 120 hpi (Figure 3 C). TaTIFY6A (TIFY domain-containing protein 6a, Gene ID: TraesCS5A02G533100) related to “Plant hormone signal transduction” pathway showed significant differential expression at 6 hpi, 12 hpi, and 24 hpi. There was an upregulation at 48 hpi, 72 hpi, and 120 hpi compared to that of 6 hpi, 12 hpi, and 24 hpi (Figure 1 D). The detailed qRT-PCR expression levels along with RNA-seq data are presented (Supplemental Table 2). The detailed KEGG enrichment pathways is presented (Supplemental Table 3).
Re-comment 1-1: I guess, you are discussing about Figure 3. Unfortunately, I do not find any supplementary files. However, the data presentation looks standard now. Please revise the Figure 3 title and describe each properly.
Response 1-1: We sincerely apologize for failing to update the relevant content in a timely manner after adjusting the order of the figures. The Figure 3B mentioned in Re-comment 1 has been renumbered as Figure 4B in the revised manuscript. To facilitate your understanding, we have inserted it into this response (Figure 1 B).
Figure 1 Cluster heatmap of DEGs (A). KEGG enrichment analysis of DEGs; only the significantly enriched pathways were showed (B).
Comment 2: However, I am not sure about the data as the control in the experiment is not clear to me. There is a chance of changes in DEGs and network analysis results once control (samples without Pst inoculation). Line 87: What did you use for the control? Line 95: Does 0 hpi mean just after inoculation?
Re-comment 2: Your answer does not clarify my question, however, I have gone through your text again (section 2.1). It means you have inoculated Pst in all plants → sprayed with water → collected samples at 0 hpi. This 0 hpi is control, right? This is a big issue, which can be a methodological error, especially for expression profiling. Usually, researchers never inoculate with pathogens for control, researchers use mock as a control. Does 0 hr mean 0 minutes? Counting proper time after inoculations is sensitive for gene expression profiling.
Response 2: I apologize for not directly explaining our inoculation details during the first revision. The comprehensive revised content is as follows:
Before conducting the experiment, we reviewed multiple references [1] (Corresponding to the references 23 in the original text) and based on the above references, we collected samples within one minute after inoculation as 0 hpi samples, designating them as the control group. The relevant description has been supplemented and revised in the original text: Line 105-Line 107.
Re-comment 2-1: It would be nice to add pictures of leaf samples after inoculation at each time points (if you have any).
Response 2-1: We apologize for failing to take photos of the samples during sample preparation. In this case, we have chosen to illustrate the mycelial changes at different time points (Figure 2 B, D, E).(Figure 2 B, D, E)
Figure 2 Phenotypes of leaves of MX169 and AM1350 after inoculation with Pst race CYR34 respectively (A). Mycelium images at different stages stained with WGA in AM1350 (B). Mycelium images at different stages stained with WGA in AM1350 (C)The mycelial lengths within AM1350 cells at different stages (D). The mycelial areas within AM1350 cells at different stages (E).
Comment 3: Another problem in this manuscript is the lack of RNA-seq data validation. Line 160: In the methodology sections, validation of the RNA-seq data is missing.
Re-comment 3: Lines 111: How did you check DNA contamination before RNA extraction? Lines 116-133: How many sequence libraries were prepared for sequencing (previously commented in Comment 9)? Lines 188-190: Which machine was used for qPCR? What is relative to TaEF-1α in Figure 2? Where are your primer list sequences for qPCR? Besides, validation of RNA-seq data with qPCR is not like Figure 6 in the manuscript. These figures have only qPCR data; RNA-seq data is missing.
Response 3: Thank you for raising these questions. We will respond to each of them individually as follows:
1, The library construction was entrusted to an RNA-seq company (Novogene Co., Ltd.). The company's returned report specified the methodology for library construction, and we briefly introduce the procedures as follows:
Before RNA extraction, agarose gel electrophoresis was performed to analyze the integrity of sample RNA and the presence of DNA contamination rRNA. No smearing was observed in the bands, indicating the absence of genomic DNA contamination. The NanoPhotometer® NP80 (Implen GmbH, Germany) was used to detect the purity of RNA, with reference to the ratios of OD260/280 and OD260/230. The total amounts and integrity of RNA were assessed using the RNA Nano 6000 Assay Kit of the Bioanalyzer 2,100 system (Agilent Technologies, CA, USA). The pure RNA ratio is 1.8-2.1. A value significantly higher than 2.1 may indicate DNA contamination. The results show that the OD260/280 and OD260/230 ratios of the RNA used for library construction fall within 1.8-2.1, confirming no DNA contamination.
2, We collected samples at 0 hpi, 6 hpi, 12 hpi, 24 hpi, 48 hpi, 72 hpi, and 120 hpi, with three biological replicates for each time point. One gene library was constructed for each sample, resulting in a total of 21 libraries built.
3, The qRT-PCR was performed on Bio-Rad CFX connect Real-Time PCR System, according to the manufacturer’s instructions.
4, TaEF-1α (Gene ID: TraesCS5D02G423400) belongs to housekeeping genes, and its encoded protein is involved in the elongation process of intracellular protein synthesis, which is essential for basic cellular life activities. The mRNA expression level of TaEF-1α is generally relatively constant across different tissues, developmental stages, or treatment conditions [1] (Corresponding to the references 28 in the original text).
5, We have supplemented the primer sequences used in qRT-PCR. The list of qRT-PCR primer sequences can be found (Supplemental Table 4).
6, We have reselected the genes involved in qRT-PCR and generated a heatmap of their corresponding RNA-seq data for comparison (Figure 3).
Figure 3 Quantitative analysis of four DEGs by qRT-PCR using cDNA templates from wheat cultivar Fielder inoculated with Pst race CYR34 at different time points (A-D).
Re-comment 3-1: Please upload the supplementary (I think it is missing).
Response 3-1: We apologize for failing to upload the attachment in the previous submission. This time, we have uploaded the attachment synchronously, and the specific information can be found in the attachment.
Comment 4: Lines 54-55: “However, a comprehensive gene expression profile of wheat in response to Pst remains unavailable ” ‒ this statement is not correct. https://doi.org/10.1128/spectrum.03774-23, https://doi.org/10.1016/j.heliyon.2022.e10951
Re-comment 4: The current sentence has remained the same as the previous.
Response 4: We apologize for the misinterpretation of your comments during the first revision.
In the original text, we stated that “However, a comprehensive gene expression profile of wheat in response to Pst remains unavailable”. After being reminded, we realized that the original expression was too one-sided and failed to fully account for the research status of gene expression profiles in wheat. Through careful study of the literature you provided, we have made the following modifications:
However, the gene expression profile of the special disease-resistant cultivar AM1350 in response to the prevalent race CYR34 of Pst remains unclear.
Re-comment 4-1: It sounds better. But I would suggest you as to write as followings
The molecular mechanisms underlying the interaction between wheat and Pst can be deciphered through global gene expression profiling approaches such as RNA sequencing (RNA-seq). RNA-seq has been widely applied to investigate the mechanisms involved in the interaction between wheat and pathogenic fungi [10,11,12,13,14]. However, resistance mechanisms can be race-specific and can be dependent on the resistance mechanisms of host plant. Despite of wide ranges of molecular studies different host plants, the resistance mechanisms of the AM1350 against CYR34 of Pst is unclear. However, the gene expression profile of the special disease-resistant cultivar AM1350 in response to the prevalent race CYR34 of Pst remains unclear [9].
Response 4-1: Thank you for your professional language revision suggestions. We have revised the original text, and the specific content is as follows:
The molecular mechanisms underlying the interaction between wheat and Pst can be deciphered through global gene expression profiling approaches such as RNA sequencing (RNA-seq). RNA-seq has been widely applied to investigate the mechanisms involved in the interaction between wheat and pathogenic fungi 1011121314. However, resistance mechanisms can be race-specific and can be dependent on the resistance mechanisms of host plant. Despite of wide ranges of molecular studies different host plants, the resistance mechanisms of the AM1350 against CYR34 of Pst is unclear.
